# Effects of Sesamin, the Major Furofuran Lignan of Sesame Oil, on the Amplitude and Gating of Voltage-Gated Na^+^ and K^+^ Currents

**DOI:** 10.3390/molecules25133062

**Published:** 2020-07-04

**Authors:** Ping-Chung Kuo, Zi-Han Kao, Shih-Wei Lee, Sheng-Nan Wu

**Affiliations:** 1School of Pharmacy, College of Medicine, National Cheng Kung University, Tainan 70101, Taiwan; z10502016@email.ncku.edu.tw; 2Department of Physiology, College of Medicine, National Cheng Kung University, Tainan 70101, Taiwan; s36084062@ncku.edu.tw (Z.-H.K.); s36081048@ncku.edu.tw (S.-W.L.); 3Institute of Basic Medical Sciences, College of Medicine, National Cheng Kung University, Tainan 70101, Taiwan; 4Department of Medical Research, China Medical University Hospital, China Medical University, Taichung 40402, Taiwan

**Keywords:** sesamin, sesamolin, Na^+^ current, M-type K^+^ current, *erg*-mediated K^+^ current, current kinetics, simulation model

## Abstract

Sesamin (SSM) and sesamolin (SesA) are the two major furofuran lignans of sesame oil and they have been previously noticed to exert various biological actions. However, their modulatory actions on different types of ionic currents in electrically excitable cells remain largely unresolved. The present experiments were undertaken to explore the possible perturbations of SSM and SesA on different types of ionic currents, e.g., voltage-gated Na^+^ currents (*I*_Na_), *erg*-mediated K^+^ currents (*I*_K(erg)_), M-type K^+^ currents (*I*_K(M)_), delayed-rectifier K^+^ currents (*I*_K(DR)_) and hyperpolarization-activated cation currents (*I*_h_) identified from pituitary tumor (GH_3_) cells. The exposure to SSM or SesA depressed the transient and late components of *I*_Na_ with different potencies. The IC_50_ value of SSM needed to lessen the peak or sustained *I*_Na_ was calculated to be 7.2 or 0.6 μM, while that of SesA was 9.8 or 2.5 μM, respectively. The dissociation constant of SSM-perturbed inhibition on *I*_Na_, based on the first-order reaction scheme, was measured to be 0.93 μM, a value very similar to the IC_50_ for its depressant action on sustained *I*_Na_. The addition of SSM was also effective at suppressing the amplitude of resurgent *I*_Na_. The addition of SSM could concentration-dependently inhibit the *I*_K(M)_ amplitude with an IC_50_ value of 4.8 μM. SSM at a concentration of 30 μM could suppress the amplitude of *I*_K(erg)_, while at 10 μM, it mildly decreased the *I*_K(DR)_ amplitude. However, the addition of neither SSM (10 μM) nor SesA (10 μM) altered the amplitude or kinetics of *I*_h_ in response to long-lasting hyperpolarization. Additionally, in this study, a modified Markovian model designed for *SCN8A*-encoded (or Na_V_1.6) channels was implemented to evaluate the plausible modifications of SSM on the gating kinetics of Na_V_ channels. The model demonstrated herein was well suited to predict that the SSM-mediated decrease in peak *I*_Na_, followed by increased current inactivation, which could largely account for its favorable decrease in the probability of the open-blocked over open state of Na_V_ channels. Collectively, our study provides evidence that highlights the notion that SSM or SesA could block multiple ion currents, such as *I*_Na_ and *I*_K(M)_, and suggests that these actions are potentially important and may participate in the functional activities of various electrically excitable cells in vivo.

## 1. Introduction

Sesame seeds and sesame oil have been widely recognized as health foods in Asian countries [1,2]. In comparison with other edible oils extracted from diverse seeds, sesame oil is extremely stable, possibly due to the effective antioxidant activities presumably attributed to its abundance of lipid-soluble furofuran lignans, such as sesamin (SSM) and sesamolin (SesA) [3,4,5,6,7]. Emerging research has previously demonstrated that SSM and SesA, the two major furofuran lignans of sesame oil, are able to suppress lipid peroxidation in erythrocytes [8], to inhibit the intestinal absorption of cholesterol and hepatic 3-hydroxy-3-methylglutaryl CoA reductase activity [9], to prevent chemically induced mammary cancer, to inhibit D^5^-desaturase and the chain elongation of C18 fatty acids [10], and to protect hypoxic neuronal and PC12 cells by suppressing ROS generation and MAPK activation [11,12,13], as well as to exhibit antihypertensive or cardioprotective effects [1,3,14]. Earlier reports have revealed that either probucol or the triterpenoid fraction of *Ganoderma*, known to possesses the antioxidant activity, could perturb the activity of ionic currents in pituitary lactotrophs [15,16]. However, whether these therapeutic lignans (e.g., SSM, SesA) can directly perturb the activity of membrane ion currents is largely uncertain.

Molecular studies of epileptogenesis have revealed that specific ion channels play essential roles in both genetic and acquired forms of epilepsy, particularly voltage-gated Na^+^ (Na_V_) channels [17,18,19,20,21]. Nine isoforms (Na_V_1.1–1.9) are found in mammalian excitable tissues, including the central nervous system, peripheral nervous system, endocrine system, skeletal muscles, and heart [22]. Moreover, several inhibitors known to preferentially block the late component of voltage-gated Na+ currents (*I*_Na_), such as ranolazine, eugenol, and perampanel, have been reported to suppress seizure activity [23,24,25]. However, whether SSM or SesA are capable of exerting any perturbation on the amplitude and gating of *I*_Na_ in response to rapid membrane depolarization remains poorly understood, though SSM was previously noted to activate transient receptor potential vanilloid type 1 in endothelial cells [26]. Alternatively, the presence of SSM has been previously revealed to suppress damage or apoptosis by streptozotocin in endocrine cells [27,28,29].

For the reasons described above, the goal of the present study was to explore whether SSM and SesA could exert any perturbations on different types of ionic currents (e.g., *I*_Na_) present in pituitary GH_3_ cells. The biophysical and pharmacological properties of ionic currents, including voltage-gated *I*_Na_, resurgent *I*_Na_ (*I*_Na(R)_), M-type K^+^ currents (*I*_K(M)_), erg-mediated K^+^ currents (*I*_K(erg)_), delayed-rectifier K^+^ currents (*I*_K(DR)_) and hyperpolarization-activated cation currents (*I*), were extensively studied in these cells. Moreover, the present work aimed to use a mathematical modeling approach for the evaluation of the perturbating actions on Na_V_-channel kinetics caused by SSM. The findings from the present observations highlight the notion that the furofuran lignans, such as SSM and SesA, are capable of perturbing the amplitude of *I*_Na_ effectively in a concentration-, time-, and state-dependent manner.

## 2. Results

### 2.1. Inhibitory Effect of Sesamin (SSM) on Voltage-Gated Na^+^ Currents (I_Na_) Identified in GH_3_ Cells

In an initial step of the experiments, we examined the effects of SSM on the amplitude and gating of *I*_Na_ in response to rapid membrane depolarization. Cells were bathed in Ca^2+^-free Tyrode’s solution containing 10 mM tetraethylammonium chloride (TEA) and the recording pipette was backfilled with a Cs^+^-containing solution. As illustrated in Figure 1A, after 1 min of exposing cells to 3 or 10 μM SSM, the amplitude in the peak and sustained component of *I*_Na_ elicited by rapid membrane depolarization from −80 mV was evidently decreased. For example, when rapid membrane depolarization from −80 to −10 mV with a duration of 40 msec was delivered (indicated in the inset of Figure 1A) to evoke *I*_Na_ [30], the addition of 3 μM SSM caused a decrease in the peak or sustained amplitude of *I*_Na_ to 139 ± 11 pA (n = 11, *P* < 0.05) or 12 ± 3 pA (n = 11, *P* < 0.05), respectively, from the control values of 248 ± 18 or 21 ± 2 pA (n = 11). After the removal of SSM, the peak and sustained amplitude returned to 232 ± 16 or 19 ± 2 pA (n = 7, *P* < 0.05).

Figure 1B illustrates that the presence of SSM can concentration-dependently depress the amplitude of peak or sustained *I*_Na_ activated during rapid membrane depolarization. The IC_50_ value needed for the SSM-perturbed decrease of peak or sustained *I*_Na_ identified in GH_3_ cells was 7.2 or 0.6 μM, respectively, the value of which was noticed to be distinct significantly between its effects on these two components. The obtained results thus demonstrate that SSM has a depressant action on the peak or sustained *I*_Na_ functionally expressed in GH_3_ cells.

### 2.2. Kinetic Constants of I_Na_ Block by SSM

During cell exposure to SSM, the *I*_Na_ in response to brief depolarization exhibited a decline in peak amplitude followed by a rise in the exponential decay of the current. For this reason, it would thus be critical to gain information about the kinetics of the SSM-induced block of these currents observed in these cells. The concentration dependence of *I*_Na_ decay (i.e., current inactivation) during a brief depolarization caused by the presence of SSM was derived and is illustrated in Figure 1C. It is important to emphasize that the effect of SSM on *I*_Na_ resulted in a concentration-dependent rise in the rate of current decay, as well as in a considerable decrease in the sustained current, notwithstanding its ineffectiveness in perturbing the initial activation phase of *I*_Na_ responding to brief depolarizing pulse. In other words, increasing the SSM concentration not only caused a reduction in the peak amplitude of *I*_Na_, but also remarkably enhanced the inactivation rate of the current in response to abrupt membrane depolarization. It stands to reason, therefore, that the inhibitory effect of SSM on *I*_Na_ identified from GH_3_ cells can be reflected with a state-dependent blocker which binds favorably to the open state of the Na_V_ channel according to a minimal binding scheme, given as follows:(1)$$C\overset{\alpha }{\underset{\beta }{⇄}}O\overset{k_{\left(+1\right)}^{*}\cdot [\mathrm{SSM}]}{\underset{k_{-1}}{⇄}}O\cdot \mathrm{SSM}$$
where α and β is the kinetic constant for the opening or closing of the Na_V_ channel, *k*_+1_^*^·[SSM] and *k*_−1_ represents the block (forward) or unblock (backward) caused by the presence of SSM, [SSM] is the blocker (i.e., SSM) concentration, and C, O, and O·SSM shown in the scheme are the closed (resting), open, and open-blocked states, respectively.

The block or unblock rate constant (i.e., *k*_+1_^*^·[SSM] and *k*_−1_) was determined from the value (i.e., τ_inact(S)_) of the slow component of the *I*_Na_ inactivation time constant during cell exposure to different SSM concentrations (Figure 1C), while SSM presence did not alter the fast component of the *I*_Na_ inactivation time course. Because a Hill coefficient of approximately 1 was obtained from the concentration–response curve, the block or unblock rate constant achieved in this study was evaluated using the formula given as follows:(2)$${\tau_{\mathrm{inact}\left(S\right)}}^{-1}=k_{+1}^{*}\cdot \left[\mathrm{SSM}\right]+k_{-1}$$

In this formula, the parameter value of *k*_+1_^*^ (the slope) and *k*_-1_ (the intercept) was calculated. As predicted from this minimum binding scheme, the relationship between 1/τ_inact(S)_ and [SSM] became linear with a correlation coefficient of 0.97 (Figure 1C). The resultant rate constant of blocking or unblocking perturbed by the addition of SSM was calculated to be 0.0449 msec^−1^μM^−1^ or 0.0415 msec^−1^, respectively; as a consequence, a value of 0.93 μM for the dissociation constant (*K*_D_ = $k_{-1}/k_{+1}^{*}\cdot \left[\mathrm{SSM}\right]$) of SSM could be achieved.

We also further examined effects of SSM on peak *I*_Na_ measured at different levels of membrane potential. As shown in Figure 1D, the experimental observations revealed that the overall current–voltage (*I-V*) relationship of peak *I*_Na_ attained between the absence and presence of 3 μM SSM did not differ noticeably, though the peak amplitude of the current measured at the level of each voltage was significantly decreased in the presence of SSM.

### 2.3. Comparison Between Effects of SSM, SesA, SSM Plus Tefluthrin, and SSM Plus Telmisartan on Peak I_Na_ Identified in GH_3_ Cells

In another experiment, we tested the effects of SSM, sesamolin (SesA), SSM plus tefluthrin, and SSM plus telmisartan on the peak amplitude of *I*_Na_ responding to rapid membrane depolarization to −10 mV from a holding potential of −80 mV. Tefluthrin, a type I pyrethroid insecticide, and telmisartan, a blocker of angiotensin II receptors, were previously demonstrated to activate *I*_Na_ directly and effectively [20,30,31,32,33]. As shown in Figure 2, SSM or SesA, at a concentration of 3 μM, produced inhibitory effects on the peak amplitude of *I*_Na_. Furthermore, in the continued presence of SSM (3 μM), the subsequent addition of either tefluthrin (10 μM) or telmisartan (10 μM) was effective in reversing the SSM-induced inhibition of peak *I*_Na_.

### 2.4. Concentration-Dependent Inhibition of I_Na_ Caused by Sesamolin (SesA)

The effects of SesA, another furofuran lignan, on *I*_Na_ in response to an abrupt depolarizing pulse were further examined and compared in this study. The concentration-dependent relationships among the inhibitory effects of SesA on the peak and sustained component of *I*_Na_ are illustrated in Figure 3. The IC_50_ value of SesA required for its effect on the peak or sustained *I*_Na_ measured at the beginning or end of a brief depolarizing pulse was calculated to be 9.8 and 2.5 μM, respectively, though these values were relatively higher than for the SSM used for the blocking of the peak or sustained *I*_Na_ in GH_3_ cells.

### 2.5. Inhibitory Effect of SSM on Resurgent I_Na_ (I_Na(R)_) in GH_3_ Cells

We next wanted to determine whether SSM exerts any effects on *I*_Na(R)_ identified from these cells [30]. The whole-cell experiments on *I*_Na(R)_ were undertaken when each cell was voltage clamped at −80 mV and a brief depolarizing step to +20 mV was delivered to activate transient *I*_Na_. The *I*_Na(R)_ upon repolarization to various potentials ranging between −50 and 0 mV was thereafter measured at the end of voltage pulses (Figure 4). The effect of SSM on *I*_Na(R)_ was examined at various membrane potentials, and the *I-V* relationship of *I*_Na(R)_ with or without the addition of SSM was constructed. The presence of SSM (1 μM) was capable of decreasing *I*_Na(R)_ with no noticeable change in its voltage dependence in GH_3_ cells, since the overall shape of the *I–V* curves for *I*_Na(R)_ appearing between the absence and presence of SSM appeared to be similar. For example, at the level of −30 mV, the exposure to 1 μM SSM resulted in a decrease in *I*_Na(R)_ amplitude from 46 ± 6 to 22 ± 5 pA (n = 8, *P* < 0.05).

### 2.6. Concentration-Dependent Inhibition of M-Type K^+^ Currents (I_K(M)_) Caused by SSM in GH_3_ Cells

The following experiments were further undertaken to determine whether the addition of SSM could exert any perturbations on the amplitude or gating of *I*_K(M)_ in these cells [34,35,36]. Cells were bathed in high-K^+^, Ca^2+^-free solution, and the recording pipette was then filled with K^+^-enriched solution. Figure 5 illustrates that the presence of SSM can result in a concentration-dependent depression in the amplitude of *I*_K(M)_ during step depolarization. The IC_50_ value needed for an SSM-perturbed decrease of *I*_K(M)_ observed in GH_3_ cells was 4.8 μM, a value noticeably higher than that used for its inhibitory effect on the late component of *I*_Na_ in response to brief depolarization.

### 2.7. Inhibitory Effect of SSM on Erg-Mediated K^+^ Current (I_K(erg)_) in GH_3_ Cells

We further explored whether SSM could perturb another types of K^+^ currents (i.e., *I*_K(erg)_ and *I*_K(DR)_) in these cells. As demonstrated previously [15,23,37], to amplify the deactivated *I*_K(erg)_, cells were bathed in high-K^+^, Ca^2+^-free solution. In this stage of the measurements, we bathed cells in high-K^+^, Ca^2+^-free solution containing 1 μM tetrodotoxin (TTX), and then filled up the electrodes by using K^+^-enriched solution. As depicted in Figure 6A,B, as cells were exposed to 30 μM SSM, the amplitude of *I*_K(erg)_ in response to negative potentials from −10 mV was evidently decreased. Figure 6B represents the average *I–V* relationship of deactivated *I*_K(erg)_ achieved in controls and during the exposure to 30 μM SSM. Therefore, SSM at a concentration higher than 30 μM can effectively depress the amplitude of *I*_K(erg)_ in GH_3_ cells.

### 2.8. Mild Inhibitory Effect of SSM on Delayed Rectifier K^+^ Currents (I_K(DR)_) in GH_3_ Cells

We next examined whether the presence of SSM is able to modify *I*_K(DR)_ present in GH_3_ cells. To achieve this goal, we bathed cells in Ca^2+^-free Tyrode’s solution which contained 1 μM TTX, and then filled up the recording electrode by using K^+^-containing solution. When cells were exposed to SSM at a concentration of 3 μM, the amplitude of *I*_K(DR)_ in response to a 1-sec step depolarization was unchanged. It is noted, however, that the *I*_K(DR)_ amplitudes responding to different levels of depolarizing command steps were decreased by the addition of 10 μM SSM, though the activation time course of *I*_K(DR)_ evoked by membrane depolarization remained unchanged. For example, as the *I*_K(DR)_ amplitude was measured at the level of +40 mV, the presence of 10 μM SSM significantly decreased the current amplitude by 23 ± 4% from 598 ± 34 to 454 ± 29 pA (n = 8, *P* < 0.05). The average *I–V* relationship of *I*_K(DR)_ obtained in the absence or presence of 10 μM was constructed and is hence illustrated in Figure 7.

### 2.9. Inability of SSM to Perturb Hyperpolarization-Activated Cation Currents (I_h_) in GH_3_ Cells

In the following experiments, we further studied whether the presence of SSM could perturb another type of inwardly directed current, i.e., *I*_h_. Cells were exposed to Ca^2+^-free Tyrode’s solution containing 1 μM TTX and the pipette was filled with K^+^-containing solution. As the hyperpolarizing command pulse from −40 to −110 mV with a duration of 2 sec was delivered, *I*_h_ with a slowly activating property was robustly evoked, as observed previously [16,38,39]. As illustrated in Figure 8, 1 min of exposure to SSM at a concentration of 10 μM was unable to modify the amplitude or gating (i.e., activation or deactivation kinetics) of *I*_h_ in response to a 2-sec hyperpolarizing pulse from −40 to −110 mV. For example, at the level of −110 mV, the *I*_h_ amplitude measured at the end of the hyperpolarizing step between the absence and presence of 10 μM SSM did not differ (388 ± 24 pA (control) versus 386 ± 26 pA (in the presence of SSM); n = 9, *P* > 0.05). Similarly, the application of 10 μM SesA had a minimal effect on *I*_h_ amplitude. However, in the continued presence of SSM (10 μM), the subsequent application of cilobradine at a concentration of 3 or 10 μM was highly effective at inhibiting the *I*_h_ amplitude in combination with a measurable slowing in the activation time course of the current. Cilobradine has recently been reported to decrease *I*_h_ amplitude, as well as to alter activation kinetics present in different types of excitable cells [39]. As such, distinguishable from its effect on *I*_Na_ or different types of K^+^ currents demonstrated above, the addition of SSM failed to alter the amplitude and kinetics of *I*_h_ identified in GH_3_ cells.

### 2.10. Simulations of SSM-Mediated Inhibition of I_Na_ Derived From a Markov State Model

To further elucidate the ionic mechanism of the inhibitory actions of SSM, a modified Markovian model used to simulate *I*_Na_ (i.e., SCN8A-encoded (or Na_V_1.6) current) was examined. The mRNA transcripts for the α-subunit of Na_V_1.1, Na_V_1.2, Na_V_1.3, and Na_V_1.6 were reported to be present in GH_3_ cells [40,41]. This model, illustrated in Figure 9A, was originally derived from Pan and Cummins [21]. The detailed meanings for the default parameters used in this model were previously elaborated [21]. Basically, the model consists of five closed states, one open state, one blocked state, and six inactivation states. As shown in Figure 9B, the inhibitory effect of SSM on simulated *I*_Na_ closely resembled the experimental observations reported above. The observations showed that the inhibitory effect of SSM at a concentration of 0.3 and 1 μM can be mimicked by an increase in Oon (i.e., transitional rate from the open to I6 state) to 3.5 and 4.6 msec^−1^ from a control value of 2.3 msec^−1^. Therefore, a progression toward the activated state became considerably raised in the presence of 0.3 or 1 μM SSM by 25% or 50%, respectively. Overall, the simulation results produced a good match to the experimental observations which disclosed that, during cell exposure to SSM (0.3 or 1 μM), the current amplitude of simulated *I*_Na_ (i.e., *SCN8A*-encoded current) in response to a brief depolarization was decreased, along with a reduction in the inactivation time constant. Additionally, on the basis of our analysis, as demonstrated in Figure 9C, when the cells were exposed to SSM, the state probability in the OB state of the channel appeared to be sensitive to a decrease to a greater extent than that in the O state. For example, as the modeled cell was exposed to 1 μM SSM, the occupancy probability in the O state mildly decreased from 0.57 to 0.52, while that in the OB state resulted in a reduction from 0.079 to 0.046.

## 3. Discussion

The principal findings obtained in the present study are as follows. First, in pituitary GH_3_ cells, SSM or SesA, known to be the therapeutic furofuran lignans of sesame oil [1,3,14], differentially and effectively inhibited the transient and late components of *I*_Na_ in a concentration-dependent manner. Second, the addition of SSM can result in a modification of the inactivation kinetics of *I*_Na_ in response to brief depolarization. Third, the presence of SSM could inhibit the amplitude of *I*_Na(R)_. Fourth, its presence concentration-dependently depressed the amplitude of *I*_K(M)_. Fifth, the presence of SSM mildly decreased the amplitude of *I*_K(erg)_ and *I*_K(M)_. Sixth, SSM itself was unable to alter the amplitude or gating of hyperpolarization-elicited *I*_h_. Seventh, according to a Markovian model designed from the SCN8A channel adopted previously [21], SSM-perturbed changes in the gating kinetics of Na_V_ channels could be predictably described from their lowering of the probability of open (O) and open-blocked (OB) states of the channel. Overall, the experimental and simulation results found here meant that the inhibition by SSM of these ion channels can be caused by one of several ionic mechanisms underlying its remarkable changes to the functional activities of different types of electrically excitable cells, supposing that similar observations can be found in vivo. To what extent these compounds have therapeutic relevance in the treatment of patients with epilepsy remains to be studied.

A noticeable feature of the block of *I*_Na_ caused by SSM in GH_3_ cells is that the initial rising phase of the current (i.e., activation time course) was unaffected. However, the inhibitory effects of SSM on *I*_Na_ are not restricted to its suppression of the peak component of the current. As was expected, increasing the SSM concentration not only decreased the peak component of *I*_Na_ responding to rapid membrane depolarization, but also accelerated the inactivation rate of the current. The SSM molecule appeared in the blocking only when the Na_V_ channel was in the open state. This feature can be incorporated into a simple kinetic scheme (i.e., closed↔open↔ open-blocked), as demonstrated in Figure 1C. As such, it is most likely that SSM or SesA preferentially binds to and blocks the open state of the Na_V_ channels.

In this study, we observed that SSM at the concentrations falling in the range between 0.1 and 0.3 μM caused little or no effect on the peak component of *I*_Na_ in response to brief membrane depolarization, whereas, at the same concentration, it effectively blocked the sustained component of *I*_Na_. In this scenario, the calculated IC_50_ value of SSM, which was required for the inhibition of sustained *I*_Na_, tends to be lower than that for its inhibitory effect on peak *I*_Na_, highly reflecting that there is a considerable and selective block of sustained *I*_Na_ caused by SSM. Meanwhile, the exposure to SSM produced a reduction in the amplitude of *I*_Na(R)_, though no change in the overall *I–V* relationship of *I*_Na(R)_ was obtained in its presence.

Sesame oil was shown to exert protective effects against cypermethrin-induced damage in genomic DNA and histopathological changes in the brain or hematotoxicities [13,43]. It was reported to prevent the deleterious effect of cypermethrin in rat liver and kidney [44,45]. The present observations showed that the SSM-mediated inhibition of *I*_Na_ could be counteracted by a further application of tefluthrin, structurally similar to cypermethrin, suggesting that pyrethroid-induced neurotoxicity could be reversed by SSM or SesA.

It should be noticed that the neurological or cardioprotective actions caused by SSM, SesA, or other structurally similar compounds, as described previously [42,46,47,48,49,50,51,52,53], can be intimately linked to their direct actions on the amplitude and gating of ion currents (e.g., *I*_Na_). Similar to the ranolazine or perampanel action on *I*_Na_ described previously [23,25], the inhibitory effect of SSM on ion currents seen herein may be responsible for its wide spectrum of effects observed in vivo [3,54]. Additionally, caution needs to be taken in the interpretation of sesame oil as a fat-soluble vehicle [55,56,57].

The present observations also revealed that SSM could decrease the amplitude of *I*_K(M)_ in GH_3_ cells with an IC_50_ of 4.8 μM. *I*_K(M)_ is biophysically characterized by a slow activation and deactivation property during step depolarization [34,35,36]. It needs to be noticed, therefore, that the inhibition of *I*_Na_ caused by SSM or SesA could be indirectly and concurrently altered by their inhibitory effects on *I*_K(M)_ observed in non-voltage-clamped cells, since the suppression of *I*_Na_ amplitude would be further exacerbated by the membrane depolarization produced by *I*_K(M)_ inhibition. In other words, the SSM-mediated inhibition of *I*_Na_ and *I*_K(M)_ studied herein likely synergistically influences the functional activities of electrically excitable cells such as pituitary lactotrophs. However, whether different lignans in dietary vegetables produce similar actions to the ones observed here still remains to be further examined.

The voltage-clamp current measurements are unable to realize the changes of the occupancy probability of each state simultaneously. In this study, the biophysical model (Figure 9A) adopted in the present study [21] tends to be based on a relatively small number of variables. However, it allowed us to virtually highlight a qualitative way of how the presence of SSM perturbs the amplitude and gating of *I*_Na_. As such, the model demonstrated herein is able to complement the experimental observations by providing insight into the gating of Na_V_ channels, which can impinge upon the electrical behavior of neurons or neuroendocrine cells. Our simulation results generated from this model support the notion that changes in the magnitude and kinetics of *I*_Na_ caused by SSM, in which varying value of Oon is the valuable parameter involved, are responsible for its actions on the functional activity of electrically excitable cells in vivo, though other additional variables also likely take part in the regulation of *I*_Na_ kinetics. Oon, appearing in the model, is the on rate of normal inactivation from the open state of the Na_V_ channel. Overall, the findings from the present simulations disclose that the decreases in both peak amplitude and the inactivation time constant of *I*_Na_, in which the SSM action is mimicked, could be a potentially important mechanism underlying the rate and pattern of repetitive firing in electrically excitable cells appearing in vivo.

## 4. Materials and Methods 

### 4.1. Chemicals and Solutions

This study used (+)-Sesamin (SSM; C_20_H_18_O_6_, [1S-(1α,3aα,4α,6aα)] -5,5’- (tetrahydro-1H,3H-furo [3–4-c]furan-1,4-diyl)*bis*-1,3-benzodioxole, https://pubchem.ncbi.nlm.nih.gov/compound/sesamin) and (+)-sesamolin (SesA; C_20_H_18_O_7_, 5-[(1S,3aR,4R,6aR)-4-(1,3-benzodioxol-5-yloxy)tetrahydro-1H,3H-furo[3,4-c]furan-1-yl]-1,3-benzodioxole, https://pubchem.ncbi.nlm.nih.gov/compound/585998). The methanol extract of sesame (*Sesamum indicum*) seeds was fractionated and purified with the assistance of conventional column chromatography to afford 29 compounds, including seven furofuran lignans. Both the extraction or fractionation of medicinal plants (i.e., sesame oil) and the basic chemical structure of SSM or SesM are illustrated in a previous paper [5]. The purity of (+)-sesamin and (+)-sesamolin, as well as the specific rotation ([a]_D_ value), are shown in the Appendix A.

Cilobradine (CIL) was obtained from Cayman (Excel Biomedical, Taipei, Taiwan), tefluthrin (Tef), tetraethylammonium chloride (TEA), and tetrodotoxin (TTX) were from Sigma-Aldrich (Merck Ltd., Taipei, Taiwan) and telmisartan (Tel) was from Tocris (Union Biomed Inc., Taipei, Taiwan). Unless otherwise stated, culture media (e.g., Ham’s F-12 medium), fetal bovine serum, horse serum, L-glutamine, and trypsin/EDTA were acquired from HyClone^TM^ (Thermo Fisher; Level Biotech, Tainan, Taiwan), while other chemicals and reagents were of analytical grade.

The bath solution (i.e., a HEPES-buffered normal Tyrode’s solution) utilized in the present study was composed of 136 mM NaCl, 5.4 mM KCl, 1.8 mM CaCl_2_, 0.53 mM MgCl_2_, 5.5 mM glucose, and 5.5 mM HEPES titrated with NaOH to pH 7.4. To measure macroscopic *I*_K(DR)_ and *I*_h_, we filled patch pipettes by using a solution containing 130 mM K-aspartate, 20 mM KCl, 1 mM KH_2_PO_4_, 1 mM MgCl_2_, 3 mM Na_2_ATP, 0.1 mM Na_2_GTP, 0.1 mM EGTA, and 5 mM HEPES adjusted with KOH to pH 7.2. To record *I*_Na_ and *I*_Na(R)_, we substituted K^+^ ions in the pipette solution inside the electrode for equmolar Cs^+^ ions and the pH value in the solution was adjusted to 7.2 with CsOH. To measure *I*_K(erg)_ and *I*_K(M)_, cells were bathed in a high-K^+^ solution containing 145 mM KCl, 0.53 mM MgCl_2_, and 5 mM HEPES-KOH buffer, pH 7.4. All solutions were prepared using deionized water from a Millipore Milli-Q purification system (ρ = 18 MΩ·cm) (Merck, Ltd., Taipei, Taiwan). The pipette solution and culture medium were filtered on the day of use with a sterile Acrodisc^®^ syringe filter with a 0.2-μm Supor^®^ membrane (Bio-Check; New Taipei City, Taiwan).

### 4.2. Cell Culture

The pituitary adenomatous cell line, GH_3_, was acquired from the Bioresource Collection and Research Center ((BCRC-60015, https://catalog.bcrc.firdi.org.tw/BcrcContent?bid=60015); Hsinchu, Taiwan). Cells were maintained in Ham’s F-12 medium supplemented with 2.5% fetal bovine serum (v/v), 15% horse serum (v/v), and 2 mM L-glutamine in a humidified environment of 5% CO_2_/95% air [31,38]. When well differentiated, GH_3_ cells were transferred to a serum- and Ca^2+^-free medium. Electrical recordings were performed 5 or 6 days after cells were cultured with 60–80% confluence.

### 4.3. Electrophysiological Measurements

On the day of each experiment, cells were dispersed with a 1% trypsin/EDTA solution and a few drops of cell suspension were rapidly placed in a custom-built recording chamber mounted on the stage of an inverted DM-IL microscope (Leica; Major Instruments, Kaohsiung, Taiwan). They were immersed at room temperature (20–25 °C) in normal Tyrode’s solution, the composition of which is elaborated above. We measured ion currents in the whole-cell model of a standard patch-clamp technique with dynamic adaptive suctioning (i.e., decremental change of suction pressure in response to a progressive increase in the electrode resistance), with the aid of an RK-400 (Bio-Logic, Claix, France) or an Axopatch-200B (Molecular Devices, Sunnyvale, CA) patch amplifier [33,38,58]. The microelectrodes used were prepared from Kimax-51 borosilicate capillaries with a 1.5-mm outer diameter (#34500; Kimble; Dogger, New Taipei City, Taiwan) by using a PP-830 vertical puller (Narishige, Taiwan Instrument, Taipei, Taiwan). The recording electrodes had their tip resistances, which ranged between 3 and 5 MΩ, as they were filled up with the different internal solutions elaborated above. During the measurements, the recorded area on the vibration-free table was shielded by using a Faraday cage (Scitech, Seoul, South Korea). The potentials were corrected for the liquid–liquid junction potential that would appear when the composition of the pipette solution remained different from that in the bath.

### 4.4. Data Recordings

The signals, composed of potential and current traces, were monitored on an HM-507 oscilloscope (Hameg, East Meadow, NY) and digitally stored online at 10 kHz in a Sony VAIO CS series laptop computer (VGN-CS110E; Kaohsiung, Taiwan) equipped with a 12-bit resolution Digidata 1440A interface (Molecular Devices). During the recordings with either analog-to-digital or digital-to-analog conversion, the latter device was controlled by pCLAMP 10.7 software (Molecular Devices) run on Microsoft Windows 10 (Redmond, WA). The laptop computer used was also put on the top of an adjustable Cookskin stand (Ningbo, Zheijiang, China) for efficient manipulation during the experiments.

### 4.5. Data Analyses

The digitized signals were examined and analyzed offline using different programs, such as pCLAMP 10.7 (Molecular Devices), 64-big OriginPro 2016 (OriginLab, Taipei, Taiwan), Prism 6 (GraphPad; SoftHome International, Taipei, Taiwan), or custom-made macros created in Microsoft Excel^®^ 2013, which was executed on Windows 10 (Redmond, WA). The concentration–response data for the inhibition of either peak and late *I*_Na_ and *I*_K(M)_ inherently in GH_3_ cells were least-squares fitted to the modified Hill equation, which can be written as follows:(3)$$\mathrm{percentage}\;\mathrm{inhibition}=\frac{E_{max}\times {\left[C\right]}^{n_{H}}}{{\left[C\right]}^{n_{H}}+IC_{50}^{n_{H}}}$$
where [C] denotes the SSM or SesA concentration given; IC_50_ and n_H_ represent the concentration required for a 50% inhibition and the Hill coefficient, respectively; and *E*_max_ is the maximal inhibition of either peak and late *I*_Na_ or *I*_K(M)_ caused by the different concentrations of SSM or SesA.

### 4.6. Statistical Analyses

Linearized or non-linearized curve fitting to the data sets was performed using either pCLAMP 10.7 (Molecule Devices), OriginPro (OriginLab), or Prism 6.0 (GraphPad). All data are presented as mean value ± SEM with sample sizes (n) indicative of the cell numbers from which the data were collected; error bars are plotted as SEM. Paired or unpaired Student’s *t*-tests were initially applied for the statistical analyses. As the statistical difference among different groups was necessarily determined, we performed either analysis of variance (ANOVA)-1 or ANOVA-2 with or without repeated measures followed by Duncan’s post hoc test. A *P*-value of < 0.05 was considered to indicate statistical difference.

### 4.7. Computer Simulations

To simulate both the increase in the degree of the *I*_Na_ inactivation rate and the decrease in the peak *I*_Na_, a modified Pan–Cummins model was mathematically constructed in the study. The state model of the *SCN8A*-encoded (or Na_V_1.6) channel which we employed in this work has been described in previous studies [21,59,60]. Such a kinetic scheme that can take into account the obtained results is described below, where C is the final closed state before opening, O is an open state, I is an inactivated state, and OB is a blocked state. The simple Scheme 1 is given as follows:

The programs designed in the present study were written in the XPP simulation package available in http://www.math.pitt.edu/~bard/xpp/xpp.html. Differential equations were solved by a fourth order Runge–Kutta algorithm. Parts of the numerical simulations were also verified with Microsoft Excel [20,61].

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
