# Peer review of "Effects of Sesamin, the Major Furofuran Lignan of Sesame Oil, on the Amplitude and Gating of Voltage-Gated Na+ and K+ Currents"

_molecules, 2020, doi:10.3390/molecules25133062_

Round 1
Reviewer 1 Report
Dear Authors,
I have read with interest your manuscript, which presents a comprehensive electrophysiological and biophysical analysis of the action of two sesame oil lignans, sesamin and sesamolin, on different subtypes of Na+ and K+ currents in GH3 cells.
Here are my observations on the manuscript:
Major points
- My major concern is on the statistical analysis of data; according to the Methods section, "Paired or unpaired Student’s t tests were initially applied for the statistical analyses; however, as the statistical difference among different groups was necessarily determined, we further implemented post-hoc Duncan multiple-range comparisons. The data that would not be deemed to comply with the assumptions of normality were also analyzed by non-parametric methods". I do not think that this is the most appropriate approach; please find below my indications on the different tests to be used for the various experiments. Related to this, please add asterisks also in plots such as the I(V) relationship in Fig. 1D, not only in histograms, and indicate the type of statistical test used in the corresponding legend.
- Figure 1: adding legends to immediately identify the differently coloured curves would greatly improve clarity.
For panel D, statistical analysis should be by ANOVA-2 for repeated measures. - Figure 2: please provide representative traces for all the treatments.
Statistical analysis should be by ANOVA-1. - Figure 3: Please add representative traces (also from the control group).
- Figure 4B: data analysis should be by ANOVA-2 for repeated measures.
- Figure 6B: data analysis should be by ANOVA-2 for repeated measures.
- Figure 7B: data analysis should be by ANOVA-2 for repeated measures.
- Figure 8B: data analysis should be by ANOVA-1.
- Do SesA or SSM have any effect on the resting membrane potential?
Minor points
- The analysis of SesA is much less detailed than that of SSM. Thus, I suggest removing the word "sesamolin" from the title, rephrasing it to "Effects of Sesamin on the Amplitude and Gating of Voltage-Gated Na+ and K+ currents".
- Line 104: "attained" should be "obtained".
- Line 213: "voltage-clamp" should be "voltage-clamped".
- Line 347, "known to be the therapeutic lignan...": please add reference(s).
- Figure 1A: the representative traces look cropped. Please adjust the y-scale to fit the whole trace.
- How long do the effects of SesA and SSM last? According to the Methods, the compounds started to act after 1 min of bath application, but it would also be interesting to know if any sort of desensitization occurs.
- What is the relevance of these results for epilepsy? This topic is cited in the Introduction (lines 60-73) as one of the reasons for this study, but nothing on this can be found in the Discussion.
Author Response
Thanks for the reviewer’s comments. The reply below follows the comments pointed out by the reviewer.
Major points
- Thanks for the comments by the reviewer. The sentence regarding non-parametric methods was removed from the revised manuscript. Moreover, as per the advice by the reviewer, Figure 1D was redone in the revised manuscript and the legend was correspondingly revised (lines 136-138).
- The labeling of Figure 1A was shown in lines 118-120 of the revised manuscript. An additional sentence was included in the revised manuscript. That is, “The statistical analyses were made by ANOVA-2 for repeated measures” (line 136-138 of the revised manuscript).
- As advised by the reviewer, Figure 2 was redone. The representative current traces were included in Figure 2A. An additional sentence (i.e., the statistical analyses were made by ANOVA-1) was included in the revised manuscript (lines 195-196).
- As advised by the reviewer, Figure 3 was redone as well. The representative current trances were incorporated into Figure 3A. The legend in Figure 3 was accordingly revised (lines 209-210).
- As per the reviewer’s advice, the sentence (i.e., “Data analysis was made by ANOVA-2 for repeated measured”) was included in the legend of Figure 4B (lines 223-224 in the revised manuscript).
- The sentence (i.e., “Data analysis was made by ANOVA-2 for repeated measured”) was included in the legend of Figure 6B (line 261-262 in the revised manuscript).
- The sentence (i.e., “Data analysis was made by ANOVA-1”) was included in the legend of Figure 8B (line 319 in the revised manuscript).
- In the present study, we intended to explore whether SSM and SesA could exert any perturbations on different types of ionic currents (e.g., INa) present in pituitary GH3 The biophysical and pharmacological properties of ionic currents, including voltage-gated INa, resurgent INa (INa(R)), M-type K+ current (IK(M)), erg-mediated K+ current (IK(erg)), delayed-rectifier K+ current (IK(DR)) and hyperpolarization-activated cation current (I), were also studied in these cells. However, to what extent SSM or SesA could have any effect on the resting membrane potential of GH3 by current-clamp conditions cells remains to be further examined.
Minor points
- As commented by the reviewer, the title was hence changed to “Effects of sesamin, the major Furofuran lignan of sesame oil, on the amplitude and gating of voltage-gated Na+ and K+ currents (lines 2-4 in the revised manuscript).
- “attained” was replaced with “obtained” (line 118 in the revised manuscript).
- “voltage-clamp” was changed to “voltage-clamped” (line 237 in the revised manuscript).
- As advised by the reviewer, the references were included in the sentence (line 371, in the revised manuscript).
- As advised by the reviewer, Figure 1D was redone.
- In our experiments, the compounds started to produce effects after 1 minute of bath application. However, we did not observe any sort of desensitization during continual 2 or 3 minutes of the presence in these tested compounds.
- As noted by the reviewer, an additional sentence was hence included in the revised manuscript. That is, “To what extent these compounds have therapeutic relevance in the treatment of patients with epilepsy remains to be studied” (lines 383-384 in the revised manuscript).
Reviewer 2 Report
Effects of sesamin and sesamolin on voltage-gated Na+ current, erg-mediated K+ current, hyperpolarization-activated cation current, M-type K+ current, delayed-rectifier K+ current, hyperpolaroztion-activated cation current were described. After considering the following points, this referee will recommend to publication.
1) The purity of (+)-sesamin and (+)-sesamolin should be shown. 2) The specific rotaion ([a]Dvalue) should be shown.
3) We can find many lignans in dietary vegetables. The reason that the authors select (+)-sesamin and (+)-sesamolin should be described.
4) The comparison of the effect with other natural products reported the same effects.
Author Response
Thanks for the reviewer’s comments, stating that the manuscript of ours could be acceptable for publication.
The reply below follow the comments pointed out by the reviewer.
- The purity of (+)-sesamin and (+)-sesamolin as well as the specific rotation ([a]D value) was shown in the Supplementary information (lines 446-447).
- Indeed, there are many lignans contained in different dietary vegetables. The main reasons why we used these two compounds are because of the fact that they are the two mayor furofuran lignans of sesame oil and have been purified (Kuo et al., J Agric Food Chem 2011;59:3214-3219).
- As pointed out by the reviewer, an additional sentence was included in the revised manuscript. That is, ”However, whether different lignans in dietary vegetables produce similar actions observed here still remains to be further examined” (lines 422-423).
Reviewer 3 Report
I would like to see information on the purity of the isolated compounds and are the results not affected by any impurities that may be present in the isolated compounds?
Author Response
Thanks for the reviewer’s comments. Supplementary information shows the purity of isolated compounds.
Round 2
Reviewer 1 Report
Dear Authors,
I read the revised version of your manuscript; most of the points I raised were addressed, but I found that the manuscript should be further improved in the following aspects:
MAJOR POINTS
- Although figure legends state that appropriate ANOVA tests have been performed, no detailed statistics summary is provided. Each sentence should be as follows: for ANOVA-1, "ANOVA-1, p<X.XXX, followed by XXXX post-hoc test, ***p<0.001, **p<0.01, *p<0.05 (select those that apply)"; for ANOVA-2 (either with or without repeated measures), p(factor 1) < X.XXX, p(factor 2) < X.XXX or p(interaction)<X.XXX, followed by XXX post hoc test, ***p<0.001, **p<0.01, *p<0.05 (select those that apply);
- Significance asterisks were added only for Fig. 1D, but they should be provided also for Fig. 4B, Fig. 6B, Fig. 7B. For histograms, horizontal lines connecting the different bars to be compared, with the corresponding asterisk(s) on top, would greatly help in visually appreciating the relevant statistical comparisons.
MINOR POINTS
- Figure 1: Yes, a key to the colors of the different traces and plots was already included in the first version of the manuscript. However, I think that a graphical color legend in the figure would help the reader to immediately understand "who is who", without the need to go back and forth from figure to legend. Please add it.
- Figure 1A: The representative traces still look cropped; please expand the y-scale (e.g., from 0 to -400 pA) to avoid the peaks to be superimposed on the x-axis.
- Figure 2A: I thank the Authors for adding the representative traces. However, I again suggest to add a graphical legend to immediately understand the differently color coded traces; I do not believe that the written indications in the legend really help the reader in interpreting the figure.
- Figure 3A: Please adjust following my comments on Figure 2A.
- Lines 500-2: This sentence does not accurately describe the actual statistical analyses performed in the text. Please correct to reflect all the statistical tests that were actually used.
Reviewer 2 Report
This referee recommends this article to the publication.
Author Response
Thanks for the reviewer’s comment, stating that the revised manuscript of ours could be acceptable for publication.